# Multifaceted Analyses of Isolated Mitochondria Establish the Anticancer Drug 2-Hydroxyoleic Acid as an Inhibitor of Substrate Oxidation and an Activator of Complex IV-Dependent State 3 Respiration

**DOI:** 10.3390/cells11030578

**Published:** 2022-02-07

**Authors:** Kumudesh Mishra, Mária Péter, Anna Maria Nardiello, Guy Keller, Victoria Llado, Paula Fernandez-Garcia, Ulf D. Kahlert, Dinorah Barasch, Ann Saada, Zsolt Török, Gábor Balogh, Pablo V. Escriba, Stefano Piotto, Or Kakhlon

**Affiliations:** 1Department of Neurology, Hadassah-Hebrew University Medical Center, Ein Kerem, Jerusalem 9112102, Israel; kumudeshmishra@gmail.com; 2Faculty of Medicine, Hebrew University of Jerusalem, Ein Kerem, Jerusalem 9112102, Israel; guy.keller@mail.huji.ac.il (G.K.); dinorah.barasch@mail.huji.ac.il (D.B.); anns@ekmd.huji.ac.il (A.S.); 3Lipodom Ltd., Dorottya Utca 35-37, 6726 Szeged, Hungary; marapeter@gmail.com (M.P.); ztorok@lipidart.com (Z.T.); baloghg@brc.hu (G.B.); 4Biological Research Centre, Institute of Biochemistry, 6726 Szeged, Hungary; 5Department of Pharmacy, University of Salerno, Via Giovanni Paolo II, 132, 84084 Fisciano, SA, Italy; annardiello@unisa.it; 6Bionam Center for Biomaterials, University of Salerno, Via Giovanni Paolo II, 132, 84084 Fisciano, SA, Italy; 7Department of Genetics, Hadassah-Hebrew University Medical Center, Ein Kerem, Jerusalem 9112102, Israel; 8Laminar Pharmaceuticals, Ctra. de Valldemossa Km. 7, 4 Parc BIT Ed. Naorte Bolque A-1°-3, 07121 Palma de Mallorca, Spain; victoria.llado@laminarpharma.com (V.L.); p.fernandez@laminarpharma.com (P.F.-G.); 9Molecular and Experimental Surgery, Clinic for General, Visceral, Vascular, and Transplant Surgery, Medical Faculty, University Hospital Magdeburg, 39120 Magdeburg, Germany; ulf.kahlert@med.ovgu.de; 10Mass Spectrometry Unit, Institute for Drug Research, School of Pharmacy, The Hebrew University of Jerusalem, Jerusalem 9112102, Israel; 11Department of Biology, University of the Balearic Islands, 07122 Palma de Mallorca, Spain

**Keywords:** 2-hydroxyoleic acid, mitochondria, molecular dynamics, respiration, glycolysis, shotgun lipidomics, bioenergetics, membrane lipid therapy

## Abstract

The synthetic fatty acid 2-hydroxyoleic acid (2OHOA) has been extensively investigated as a cancer therapy mainly based on its regulation of membrane lipid composition and structure, activating various cell fate pathways. We discovered, additionally, that 2OHOA can uncouple oxidative phosphorylation, but this has never been demonstrated mechanistically. Here, we explored the effect of 2OHOA on mitochondria isolated by ultracentrifugation from U118MG glioblastoma cells. Mitochondria were analyzed by shotgun lipidomics, molecular dynamic simulations, spectrophotometric assays for determining respiratory complex activity, mass spectrometry for assessing beta oxidation and Seahorse technology for bioenergetic profiling. We showed that the main impact of 2OHOA on mitochondrial lipids is their hydroxylation, demonstrated by simulations to decrease co-enzyme Q diffusion in the liquid disordered membranes embedding respiratory complexes. This decreased co-enzyme Q diffusion can explain the inhibition of disjointly measured complexes I–III activity. However, it doesn’t explain how 2OHOA increases complex IV and state 3 respiration in intact mitochondria. This increased respiration probably allows mitochondrial oxidative phosphorylation to maintain ATP production against the 2OHOA-mediated inhibition of glycolytic ATP production. This work correlates 2OHOA function with its modulation of mitochondrial lipid composition, reflecting both 2OHOA anticancer activity and adaptation to it by enhancement of state 3 respiration.

## 1. Introduction

Changes in the type and/or amount of membrane lipids can be associated with different cancer states and might affect signaling cascades involved in the pathogenesis and, conversely, the therapy of cancer. Therefore, a novel therapeutic approach, called membrane lipid therapy (MLT), or melitherapy, focuses on the development of chemotherapeutic agents, which target membrane lipids, modulate their composition and, consequently, biophysical properties and protein interactions [1,2,3,4]. A protagonist of melitherapy is the synthetic fatty acid α-hydroxy-9-cis-octadecanoic acid (2OHOA). 2OHOA is an orally bioavailable nontoxic derivative of oleic acid designed to treat human cancer [5] and demonstrated to be efficacious and safe in clinical trials (e.g., ClinicalTrials.gov accessed on 4 February 2022, identifiers NCT01792310 and NCT03366480). 2OHOA’s mechanism of action is based on modulation of both membrane lipid composition/structure and expression of the membrane-embedded sphingomyelin synthase 1 [6,7,8,9,10]. These changes lead to changes in the expression, localization and activity of proteins implicated in controlling proliferation, differentiation and senescence. For instance, protein kinase C (PKC)-dependent differentiation and growth arrest signaling entails its membrane anchoring. The latter can be accomplished by a 2OHOA-mediated increase in the extent of non-lamellar phase and/or reduction in lateral surface pressure of the lipid bilayer [11,12,13,14], enabling hydrophobic domains of peripheral proteins to interact with deep hydrophobic regions of the membrane. Changes in membrane composition are also associated with signaling initiation. Thus, transfer of the polar head of phosphatidyl choline (PC) to ceramide by 2OHOA can increase the levels of diacylglycerol (DAG) and, through that, the binding of DAG-interacting PKC [15] to the membrane. Inhibition of proliferation signaling in cancer cells can also be facilitated by 2OHOA-dependent changes in membrane composition. For instance, by reducing the levels of small polar head lipids such as phosphtidylethanolamine (PE), 2OHOA can attenuate the docking of bulky oncoprotein lipid anchors, such as the isoprenylated Ras [5].

A major effect of 2OHOA on membrane lipid composition is its incorporation into pre-existing lipids [16,17]. Hydroxylated lipids were reported to activate sphingomyelin synthase 1 expression and elevate sphingomyelin levels [7,11]. However, while this effect was challenged [18], another effect of hydroxylated fatty acids is their ability to depolarize the inner mitochondrial membrane (IMM) [19,20]. This effect prompted us to investigate a new modality of cancer-selective toxicity of 2OHOA: its capacity to act as an OxPhos uncoupler [21]. This work has demonstrated that beyond acting as a cancer cell-selective OxPhos uncoupler, 2OHOA also compromises the capacity of cancer cells to compensate for the OxPhos deficiency by activating glycolysis. Here, we explored the mechanism by which 2OHOA compromises mitochondrial function. Our studies are based on the U118MG glioblastoma cell line and mitochondria derived therefrom, as most of the anticancer effects of 2OHOA were discovered in this glioblastoma cell line [4,5,7]. Lipidomics analyses of mitochondria isolated from U118MG cells show that 2OHOA treatment led to hydroxylation of their lipids. As revealed by molecular dynamics, these lipid hydroxylations decreased co-enzyme Q (CoQ) diffusion in the liquid disordered membrane domain, such as the one embedding respiratory complexes. This reduced CoQ diffusion is consistent with the inhibition of complex I–III activities when the complexes are measured separately and not as part of the entire ETC. However, the stalled CoQ membrane diffusion cannot explain the 2OHOA-mediated facilitation of state 3 respiration and complex IV electron transfer as measured within the ETC in intact mitochondria. At the same time, 2OHOA inhibited β oxidation and glycolysis, suggesting that the enhanced/compensatory complex IV activity and state 3 respiration can sustain cancer cell viability in 2OHOA-treated cells and might, therefore, be a relevant therapeutic target for boosting 2OHOA cancer cell toxicity. This work is the first study demonstrating that melitherapy can modulate membrane composition and, consequentially, activity in an isolated organelle.

## 2. Materials and Methods

### 2.1. Western Immunoblotting

Protein factions dissolved in sample buffer with a reducing agent (Life Technology, Carlsbad, CA, USA) were denatured by heating (10 min, 70 °C). We loaded the samples (20 μg protein) on 4%–12% gradient gels (Invitrogen, Waltham, MA, USA) with PrecisionPlus^®^ Pre-stained Protein Standard protein ladder (Bio-Rad, Hercules, CA, USA) in MES SDS running buffer (Life Technology). Samples were run for 60 min at 150V in the Mini Gel Tank 21 apparatus (Life Technology). Protein transfer onto nitrocellulose membrane was done using the iBlot^®^ 2 Gel Transfer Device (Life Technology). Membranes were blocked for 1 h with 5% BSA (bovine serum albumin) in Tris-buffered saline with Tween 20 (TBST, pH = 8). Blots were then probed at 4 °C overnight with mouse primary antibodies against Tomm20 (AbCam, Cambridge, UK, ab186735), alkaline phosphatase (AbCam, ab133602) and the KDEL receptor (Santa Cruz, Dallas, TX, USA, sc-57347) to detect the mitochondrial, plasma membrane, and endoplasmic reticulum fractions, respectively. Subsequently, blots were probed with HRP-conjugated secondary antibodies (AbCam, ab205718 for Tomm20 and alkaline phosphatase, and ab205719 for KDEL receptor) diluted in blocking buffer, for 1h at room temperature. Protein bands were detected by the ECL femto-kit (CYANAGEN, Bologna, Italy).

### 2.2. Lipidomics

20 × 10^6^ U118MG cells were treated with 200 µM 2OHOA for 48 h and then mitochondria were isolated by ultracentrifugation according to [22]. Lipids were extracted using a modified version of the Folch extraction protocol, as described in [23]. Electrospray ionization mass spectrometric analyses were performed on an Orbitrap Fusion Lumos instrument (Thermo Fisher Scientific, Bremen, Germany) equipped with a TriVersa NanoMate robotic nanoflow ion source (Advion BioSciences, Ithaca, NY, USA), as detailed in [24,25]. The 5 µL lipid extract was diluted with 145 µL infusion solvent mixture (chloroform:methanol:iso-propanol 1:2:1, by vol.) containing an internal standard mix [25]. Next, the mixture was halved, and 5% dimethylformamide (additive for the negative ion mode) or 4 mM ammonium chloride (additive for the positive ion mode) was added to the split sample halves. Lipids were identified by the LipidXplorer software [26]. Data files generated by LipidXplorer queries were further processed by in-house Excel macros.

Lipid classes and species were annotated according to the classification systems for lipids [27]. In sum formulas for glycerolipids, e.g., PC(34:1), the total numbers of carbons followed by double bonds for all chains are indicated. For sphingolipids, the sum formula, e.g., Cer(34:1:2) (ceramide(34:1:2), specifies first the total number of carbons in the long-chain base and the fatty acid (FA) moiety, then the sum of double bonds in the long-chain base and the FA moiety, followed by the sum of hydroxyl groups in the long-chain base and the FA moiety.

Lipidomic data are presented as mean ± SEM. Student’s *t*-tests were performed for pairwise comparisons; significance was determined according to Storey and Tibshirani [28] and was accepted for *p* < 0.05 corresponding to a false discovery rate < 0.015. Multivariate statistical analysis of lipidomic datasets was performed using MetaboAnalyst [29].

### 2.3. Molecular Dynamics Simulations

Molecular dynamics (MD) simulations were performed on palmitoyloleoylphosphatidylcholine (POPC) (100%) and POPC/2OH-POPC (50%–50%) as liquid disordered (**Ld**) membrane models and on sphingomyelin/cholesterol (SM/CHOL) (60%–40%) and SM/2OH-SM/CHOL (30%–30%–40%) as liquid ordered (**Lo**) membrane models. Membranes were constructed using the Membrane Builder tool from CHARMM-GUI. Yasara Structure v 21.6.17 software was used for membrane simulation. The four membrane models were simulated in the presence of ubiquinone and ubiquinol (target). In each membrane model, 4 copies of the particular target molecule were added. For each system, 4 copies of the molecule of interest were placed inside the membrane. The force-field used is AMBER14 under NPT (standard pressure and temperature) conditions, with the Berendsen thermostat. The applied cut-off is 8 Å with Particle Mesh Ewald (PME). All systems are fully hydrated, respecting the water density at 0.997 g/mL, and the simulation cell was neutralized with NaCl at a final concentration of 0.9%. The MD simulations lasted 30 ns. The averaged diffusion of the molecules in membranes was calculated during the last 20 ns of simulation.

The diffusion coefficients were calculated according to the Einstein equation:(1)$$\langle {\left|r\left(t\right)-r\left(0\right)\right|}^{2}\rangle =6Dt$$
where *r* is the position vector of the particle, *t* is time and *D* is the diffusion coefficient. The movement of CoQ atoms in the benzoquinone ring, and in the end of the poly-isoprenoid tail, was followed, and diffusivity coefficients were calculated as the average between different atoms of the molecule and between different copies of the molecule in the membrane.

### 2.4. Determining the Activity of Mitochondrial Enzymes

Mitochondrial enzyme activities were determined by spectrophotometric methods, as described in [30,31]. In total, 10^6^ U118MG cells per run were homogenized and spectrophotometric determination of citrate synthase, complex I, complexes II + II, and complex IV in sonicated (permeabilized) cell homogenates followed.

### 2.5. Flow Cytometry

A total of 10^6^ U118MG cells, 2OHOA-treated or control, were centrifuged and their medium was replaced with 200 μL serum-free medium and stained with 0.4 μL Mitotracker Deep Red. Cells were incubated with the dye for 30 min at 37 °C and then spun down and the medium was replaced with 1 mL PBS. Following washing, the cells were fixed with 1% paraformaldehyde in PBS and incubated with the fixing agent for 30 min at room temperature. The cells were then washed, resuspended in 1 mL PBS and analyzed for mitotracker staining (mitochondrial mass) by the Amnis^®^ ImageStream^®X^ Mk II high-resolution microscopy and flow cytometry system using the 642 nm laser line and the 702/85 BP filter. For determining glycolysis based on uptake of the fluorescent D-glucose analog NBDG, 2OHOA-treated and control U118MG cells were analyzed for NBDG fluorescence (488 nm laser line, 533/55 BP filter) in the BD LSRII flow cytometer according to the instructions of the kit manufacturer (AbCam).

### 2.6. Evaluating β-Oxidation by Mass Spectrometry

Dimethylheptanoyl-CoA (DMH) was determined by tandem liquid chromatography mass spectrometry (LC-MS/MS) using 21:0 Coenzyme A as a non-endogenous internal standard (IS). Prior to the experiment, a calibration curve of DMH (0–5 μg/mL) was established. U118MG cells were grown for 72 h with no addition, with phytanic acid (0.2 mM dissolved in 0.05 M defatted BSA (Merck, Kenilworth, NJ, USA, A4612) supplemented with 0.4 mM L-Carnitine), or with phytanic acid and 200 μM 2OHOA added in the last 48 h. Cells were then trypsinized, washed twice in serum-free DMEM, resuspended in 200 μL DDW and divided in 5 × 33 μL aliquots, each supplemented with 167 μL of different concentrations of DMH dissolved in IS solution. These solutions were left for 10 min at room temperature for extraction and cleared by sonication and separation of insoluble pellet using centrifugation at 3000 rpm. DMH was separated by UHPLC, ionized by electrospray and fragmented and analyzed by LC-MS/MS. LC-MS/MS analyses were conducted on a Sciex (Framingham, MA, USA) Triple Quad™ 5500 mass spectrometer coupled with a Shimadzu (Kyoto, Japan) UHPLC System, as detailed in Appendix A.

### 2.7. Bioenergetic Studies

Mitochondria were isolated from 2OHOA treated and control U118MG cells by ultracentrifugation as described [22]. Isolated mitochondria were then seeded at 2 μg/well in a Seahorse XF96 well plate. Mitochondrial coupling was assayed by the sequential addition of ADP (4 mM final), oligomycin (2.5 μg/mL final), FCCP (4 μM final), and Antimycin A (4 μM final) to state 2 respiring mitochondria supplied with rotenone and succinate as a respiratory substrate. Mitochondrial electron flow was assayed by the sequential addition of rotenone, succinate, Antimycin A and the complex IV substrate Ascorbate/*N*,*N*,*N*′,*N*′-Tetramethyl-p-phenylenediamine dihydrochloride (TMPD) to slightly depolarized (4 μM FCCP) mitochondria provided with pyruvate and malate as substrates. Assays were performed as detailed in Agilent’s Seahorse application note (https://www.agilent.com/cs/library/applications/5991-7145EN.pdf, accessed on 4 February 2022). Glycolytic and mitochondrial ATP production was determined by Agilent’s Seahorse machine and the real-time ATP rate assay kit following the manufacturer’s instructions. Glycolysis was estimated by Agilent’s Seahorse machine using the Glyco Stress kit following the manufacturer’s instructions.

### 2.8. Statistical Analysis

All experiments were repeated at least three times. Differences between means were tested by the Student’s *t*-test (all figures). Dimensionality reduction and separation between the lipidomic profiles of 2OHOA treated and untreated mitochondria was demonstrated by principal component analysis (PCA). We used two-way ANOVA to test the effects of both lipid composition (ordered or disordered, hydroxylated or non-hydroxylated) and redox status of CoQ (ubiquinol vs. ubiquinone) on the membrane diffusivity of CoQ. One-Way ANOVA with Sidak’s post hoc correction for multiple comparisons was used to test the statistical significance of changes in mitochondrial and glycolytic ATP production rates.

## 3. Results

### 3.1. Lipidomics Analysis of Isolated Mitochondria

Mitochondria were isolated from the U118MG human glioblastoma cell line. High mitochondrial enrichment of the mitochondrial fraction was confirmed at the protein and membrane lipid levels. At the protein level, we demonstrated the presence of the mitochondrial marker Tom20, and the absence of KDEL receptor and alkaline phosphatase, as respective markers of microsomes and plasma membrane, which can usually contaminate the mitochondrial fraction (Figure 1A).

To identify the effect of 2OHOA treatment on the mitochondrial lipidome of U118MG cells, we performed quantitative mass spectrometry-based shotgun lipidomics. At the membrane lipid level, the mitochondrial fraction, compared to the whole-cell inputs, manifests enrichment in the mitochondrial membrane markers cardiolipin and phosphatidylglycerol and depletion in the plasma membrane markers phosphatidylserine and sphingomyelin (Figure 1B).

2OHOA treatment resulted in a very clear separation of non-treated vs. treated mitochondrial lipidomes as assessed by principal component analysis (Figure 2A). The main effect of 2OHOA on mitochondrial membranes was its incorporation into glycerophospholipids (Figure 2B and Appendix A). C2-hydroxylated and odd-chain species indicate the esterification of 2OHOA in its original form [16] and after conversion to 17:1 fatty acid, respectively. This was especially pronounced in mitochondrial phosphatidylcholine and cardiolipin, which incorporated 2OHOA and 17:1 fatty acyl moieties at similar proportions. As a result, the ratio of hydroxylated lipid species accounted for 50%–60% of the corresponding non-2OHOA-generated species following 2OHOA treatment (Figure 2C). Another significant effect of 2OHOA on mitochondrial membranes is a decrease in the level of acylcarnitines (Figure 2B), which might inhibit mitochondrial β-oxidation for which acylcarnitines are rate-limiting.

### 3.2. Molecular Dynamic Simulation of the Effect of 2OHOA on Mitochondrial Membranes

We performed molecular dynamics simulations on **Lo** (SM/CHOL (60%–40%) and SM/2OHSM/CHOL (30%–30%–40%)) and **Ld** (POPC (100%) and POPC/2OH-POPC (50%–50%)) membrane models to simulate and compare the effect of 2OHOA on the diffusion of ubiquinone and ubiquinol as the electron carrier prosthetic group in the mitochondrial inner membrane (see Figure 3A for a molecular dynamics snapshot of a membrane-embedded ubiquinol). As seen in Figure 3B, ubiquinone always has greater diffusivity in the membrane than its reduced form ubiquinol. The partial hydroxylation of the aliphatic chains of SM and PC has a diffusivity-reducing effect. This is reasonable considering the increased likelihood of establishing H-bonds (Figure 3C,D). However, in more ordered membranes (SM-CHOL, Figure 3D), the introduction of hydroxyl groups only negligibly reduces CoQ mobility. It is known [32] that in complex I, NADH is oxidized at the top of the hydrophilic domain by a flavin mononucleotide, and that electrons are then transferred along an iron-sulfur cluster chain to the ubiquinone. The effect of membrane hydroxylation may also have the effect of slowing down the activity of complex I.

The 2OHOA-mediated hydroxylation decreases CoQ (both ubiquinone and ubiquinol) diffusion in **Ld** membranes, and as OxPhos complexes I–III in mitochondria are in **Ld** (not in rafts, as opposed to the plasma membrane) [33]; 2OHOA inhibits the CoQ-dependent electron transfer in respiratory complexes I–III. Conversely, in complex IV (cytochrome c oxidase), which is not dependent on CoQ, electron transfer is accelerated due to the depolarizing effect of 2OHOA [21].

### 3.3. 2OHOA Influence on Substrate Availability and Substrate Oxidation

We have previously demonstrated that 2OHOA is an OxPhos uncoupler [21]. To better understand the bioenergetic effects of 2OHOA, we also tested its effects on respiratory complex activity using spectrophotometric methods [34,35]. As our results (Figure 4A) show, 2OHOA treatment increased complex IV activity, but, surprisingly, also down modulated the activity of complexes I, II, and III, consistent with the reduction of the diffusion rate of CoQ as an electron carrier from complexes I and III and II and III. 2OHOA also led to an overall increase in mitochondrial mass, as shown by the increase in citrate synthase activity, as a marker of intact mitochondria (Figure 4A), and by staining with the potential-independent mitochondrial marker Mitotracker and flow cytometry (Figure 4B).

We also tested whether 2OHOA can inhibit mitochondrial β oxidation of lipids by a possible competitive inhibition of upstream peroxisomal α oxidation (2OHOA can undergo α oxidation and is thus converted to C17:1 (PVE, patent #PCT/ES2021/070068)). To that end, we quantified, by mass spectrometry, the extent of β oxidation of the branched fatty acids phytanic acid. Phytanic acid first undergoes α-oxidation in the peroxisome and is eventually converted to pristanic acid, which is converted to the 11 carbon intermediate 4,8-dimethylnonanoyl-CoA (C11-CoA), which is carnitine-esterified and translocated to the mitochondria for complete β-oxidation [36] (Figure 5A). 2OHOA is structurally similar to one of the fatty acids in these cascade, 2-hydroxyphytanoyl-CoA, which is also hydroxylated at the α carbon. Therefore, we hypothesized that 2OHOA might competitively inhibit fatty acid β-oxidation at this upstream peroxisomal step, leading to a restriction in the availability of the β-oxidation substrate C11-CoA. To test this hypothesis, we fed U118MG cells with phytanic acid as a precursor to the mitochondrial β oxidation biomarker 2,6-dimethylheptanoyl CoA (DMH, C9-CoA). We tested the rate of formation of this biomarker with and without pre-treatment with 2OHOA. Our results (Figure 5B,C) demonstrate a clear inhibition of thus measured β oxidation by 2OHOA.

### 3.4. Modulation of Electron Transfer and Coupling by 2OHOA

We tested the effect of 2OHOA on bioenergetics in mitochondria isolated from U118MG cells treated with vehicle control, or 200 μM 2OHOA for 48 h. We conducted two types of experiments: The coupling experiment (Figure 6A) examines the extent to which mitochondrial electron transport (ETC) is coupled to oxidative phosphorylation (OxPhos). Beginning with a state 2 substrate (succinate)-limited respiration, ADP is added to induce state 3 respiration limited by ETC only. Oligomycin is then added to inhibit respiration to state 4, where OxPhos is blocked. The respiratory control ratio (RCR, state 3/state 4) can thus be calculated. The addition of the uncoupler FCCP enables the determination of the maximal respiratory capacity of the mitochondria. The complex III blocker antimycin A is then added to obtain background respiration signal where mitochondrial respiration is blocked. As shown in Figure 6A, 2OHOA treatment led, expectedly [21], to an uncoupling effect and increased oxygen consumption rate (OCR), as compared to untreated cells, only in response to ADP and the uncoupler FCCP, while not increasing OCR in response to oligomycin supplementation, suggesting a lack of effect of 2OHOA on complex V.

We also tested the same effects of 2OHOA treatment on mitochondrial electron transfer (Figure 6B,C). This experiment examined sequential electron transfer through the different ETC complexes and can, thus, identify which of them causes mitochondrial dysfunction if observed. As our results show, following the addition of pyruvate and malate as precursors for NADH formed in the tricarboxylic acid (TCA) cycle, the electron transfer rate was increased over time in a rotenone (complex I inhibitor)-resistant manner. Supplementation of the complex II substrate succinate differentially enhanced respiration, or OCR in 2OHOA treated cells (Figure 6B, a blowup of part of Figure 6C to show the ifferences). While mitochondrial respiration was equally inhibited by the complex III inhibitor antimycin A, complex IV activity, measured following supplementation of its substrate TMPD/Ascorbate, was increased by 2OHOA (Figure 6C). There is an apparent discrepancy between the electron transport (Figure 6B,C) and spectrophotometric (Figure 4A) complex activities—i.e., according to Figure 6B,C, 2OHOA doesn’t change complex I activity and even increases complex II activity, whereas, according to Figure 4A, 2OHOA inhibits complexes I and (II + III) activities. This discrepancy probably results from the fact that spectrophotometric measurements are conducted in permeabilized membranes, not confined within a closed compartment, where the activity of every complex is measured independently [34].

### 3.5. 2OHOA Effect on ATP Production and Glycolysis

As we expected 2OHOA to cause a profound modulation of cell metabolism and ATP production routes, we decided to investigate the cellular sources of ATP production using Agilent’s Seahorse machine. Our results (Figure 7A) show that at the cell level, the dependence of 2OHOA-treated cells on OxPhos for ATP is almost absolute. While acute on assay supplementation of 2OHOA leaves glycolytic ATP production intact and compromises mitochondrial ATP production, chronic (48 h) exposure to the same concentration of 2OHOA (200 μM) completely compromises glycolysis as a source for ATP, leaving behind only mitochondrial respiration as a source of ATP. Reduction of glycolysis by 2OHOA treatment (see also [21]) is also manifested as compromised uptake of the glucose analog NBDG (Figure 7B) and as reduced extracellular acidification rate (ECAR, Seahorse XF Glyco Stress kit, Figure 7C) following supplementation of glucose (basal glycolysis) and oligomycin (maximal glycolysis). Therefore, while compromised, OxPhos seems to be almost the sole energy source in 2OHOA-treated U118MG cells.

## 4. Discussion

We previously showed [21] that 2OHOA can increase leak and reduce coupling efficiency in cancer cell lines, while not affecting leak and coupling efficiency in non-cancerous cells. In addition, in U87-MG glioblastoma cells, we demonstrated that the modular kinetics of 2OHOA correspond only to those of an uncoupling agent and not to other OxPhos modulators. This work investigated the molecular mechanism of action by which 2OHOA modulates mitochondrial respiration. Our general assumption was that since 2OHOA depolarized the IMM, it was expected to facilitate electron flux through the respiratory chain, which would also account for its documented capacity to lower mitochondrial ROS production [21]. We further assumed that this change is mediated by modifications of mitochondrial lipid composition and consequently the biophysical properties of its lipid bilayers. This assumption is supported by the finding that also high charge/high energy ions affect complex I and mitochondrial activities through modulation of mitochondrial lipid composition [37]. Our lipidomics analysis showed that the main change in mitochondrial lipid composition caused by 2OHOA was its incorporation to pre-existing membrane lipids, manifested by lipid hydroxylation. Therefore, we tested the effect of lipid hydroxylation on electron transport in the IMM using molecular dynamics. We discovered that phospholipid hydroxylation had a significant effect on the diffusion of the electron shuttle CoQ. Hydroxylation decreased CoQ diffusion (benzoquinone ring and poly-isoprenoid tail) in the **Ld** membrane subdomain. However, it had mixed or no effects on CoQ diffusion in the **Lo** subdomain (Figure 3). As respiratory complexes in the IMM are in the **Ld** domain, not in **Lo** or rafts, absent in this membrane [33], our results suggest that 2OHOA-mediated hydroxylation can inhibit respiratory complexes. Indeed, our spectrophotometric measurements (Figure 4) show that 2OHOA inhibited the activity of complexes I, II and III. As electron transfer between complex I and complex II to complex III is mediated by CoQ, these results are consistent with the predicted inhibition of CoQ electron transfer by lipid hydroxylation in the **Ld** inner mitochondrial membrane.

Measuring mitochondrial respiration in isolated mitochondria provided additional information since membrane integrity is preserved and since the activity of each complex is measured as part of ETC complexes working together, coupled to the generation of the proton gradient and ATP synthesis. Interestingly, provided with pyruvate and malate as fuel for respiration, isolated mitochondria from control and 2OHOA-treated cells showed similar respiration rates (Figure 6A). However, provided with succinate as fuel, mitochondria from 2OHOA-treated cells manifested a considerably higher respiration rate (Figure 6A (first and second time points)). This finding suggests that 2OHOA did not influence the extent to which NADH, as a complex I substrate, was generated from pyruvate and malate in the tricarboxylic acid (TCA) cycle, but rather influenced respiration rate by modifying ETC kinetics within the IMM. When succinate is provided as a respiratory substrate, respiration depends only on IMM components. A similar observation was made when mitochondria from frozen tissue, whose membrane integrity is compromised, causing leakage of TCA enzymes, were compared to fresh mitochondria: TCA-dependent pyruvate/malate fueling supported respiration only in fresh and not in frozen mitochondria, which required either NADH as a direct complex I substrate, or succinate, bypassing TCA dependence [38]. This pyruvate-malate-driven respiration was, to some extent, rotenone-resistant, as it increased with time, suggesting partial inhibition of rotenone, or the presence of a rotenone-resistant dehydrogenase. Importantly, the electron transfer rate shown in Figure 6B took place in the presence of 4 μM FCCP as an uncoupler, so that it wasn’t constrained by a proton gradient, nor required ADP and ATP synthase activity to allow electron transfer by consumption of the proton gradient.

A more pronounced effect of 2OHOA on mitochondrial respiration can be observed if we consider its impact on coupled respiration (Figure 6A). The coupling assay in isolated mitochondria shows that succinate-based respiration rate is enhanced by 2OHOA, as expected from its uncoupling effect in glioblastoma cells [21]. In contrast to isolated complex activity, which, not being in a closed compartment, precludes the establishment of a proton gradient, respiration in the coupling assay is controlled by ATP turnover and the proton gradient. 2OHOA-treated cells are more affected by these controls. This can be demonstrated by the ADP-dependent stimulation of state 3 respiration, which occurs in both 2OHOA-treated and untreated cells, but is more prominent in the treated ones, and by the increased sensitivity of OCR in 2OHOA-treated cells to the ATPase inhibitor oligomycin. Apparently, the inhibition of complex II + III activity by 2OHOA (Figure 4A), measured using supraphysiological concentrations of the electron acceptor oxidized cytochrome c and limited only by intra/inter-complex electron transfer rate [34,35,39], does not reflect physiological complexes II + III activities within proton gradient-limiting conditions. We suggest that, under these physiological conditions, the 2OHOA-stimulated complex IV activity (Figure 4A and Figure 6) serves as an energetic sink, especially given that its reduction potential is the highest in the ETC.

Moreover, the mitochondrial membrane lipidomic modifications induced by 2OHOA are in line with complex activation: Depletion of PE, which was increased by 2OHOA in U118MG cells (Figure 2B), caused complex IV inhibition in yeast [40]. Moreover, CL was shown to stabilize complex III to IV interaction and the ATP/(ADP + Pi) exchange system, which couples ATP turnover to ADP-dependent state 3 respiration [41]. While normal CL was not significantly up-modulated by 2OHOA, the synthetic lipid-generated hydroxylated CL increased the total amount of CL by 50% (Figure 2C). Another possible electron transfer facilitating effect of increased CL is stimulation of proton conductance through the ATPase F0 subunit, decreased in cancer by mitochondrial CL reduction of the mitochondrial F0F1-ATPase [42,43].

It was interesting to discover that in addition to its ETC-related effects, 2OHOA also affected mitochondrial respiration at the level of mitochondrial fatty acid oxidation. While mixed inhibitory and activating effects were observed for ETC-related effects, 2OHOA demonstrated only inhibitory effects on β oxidation. Using phytanic acid as a precursor for α and subsequent β oxidation, we show that 2OHOA reduced the generation of the β oxidation product DMH. Based on the structural similarity between 2OHOA and one of the peroxisomal oxidation intermediates 2-hydroxyphytanoyl-CoA, we propose that this inhibitory effect is due to competitive inhibition of the peroxisomal enzyme 2-hydroxyphytanoyl-CoA lyase (Figure 5A). However, at any rate, the demonstrated inhibition of the α and β oxidation cascade in peroxisomes and mitochondria (Figure 7) probably explains the reduction in the level of acylcarnitines downstream of peroxisomal α oxidation, as also observed independently in our lipidomics results (Figure 2). These results suggest that 2OHOA inhibits β oxidation, which is rate-limited by acylcarnitine mitochondrial uptake.

The mechanistic confirmation of 2OHOA as an OxPhos uncoupler probably has broader metabolic implications. An important outcome of the facilitation of respiratory electron transfer rate is increased generation of NAD^+^, or reduction of the NADH/NAD^+^ ratio, by IMM depolarization [44]. NAD^+^ is a key metabolite that can govern the main cellular metabolic pathways and is even claimed to outweigh ATP demand as the driver of aerobic glycolysis in cancer [44]. We can conjecture that if, as expected, NAD^+^ is indeed increased by 2OHOA, the added NAD^+^ would not be diverted to glycolysis, which is reduced by 2OHOA (Figure 7), and probably also not to TCA, which was not affected by 2OHOA (Figure 6). We propose that the putative 2OHOA-generated NAD^+^ surplus would be diverted to other oxidation reactions in glioma cells, such as mitochondrial biogenesis [45,46] induced by 2OHOA (Figure 4).

Lastly, the specific effects of 2OHOA on mitochondrial and glycolytic functions suggests that its therapeutic potency might be correlated with the metabolic stratification of glioblastoma tumors, recently highlighted as a critical, understudied factor in therapy design [47]. For instance, Garofano et al. [48] describe the metabolic subtypes of glioblastoma characterized by increased dependence on OxPhos, or on aerobic glycolysis. Such tumors might be distinctively vulnerable to 2OHOA treatment. Moreover, glioblastoma subtypes characterized by Warburg effect reversal (increased reliance on OxPhos for proliferation), such as Aurora kinase A-dependent [49], might also be more susceptible to 2OHOA given their increased reliance on fatty acid oxidation, which, as we show here, is specifically targeted by 2OHOA. The specific bioenergetic effects of 2OHOA discovered here could promote its personalized use as a second line in intractable radio- and chemo-therapy-resistant glioblastoma tumors.

## 5. Conclusions

This work is a mechanistic investigation of the effect of the anti-cancer drug 2OHOA on mitochondrial function and cell metabolism. We show that, applied to the glioma cell line U118MG, 2OHOA acts as a multifaceted effector of bioenergetic and metabolic pathways. 2OHOA substantially modifies mitochondrial lipid composition leading to lipid hydroxylation, which could explain the inhibition of the innate activity of respiratory complexes I–III and the stimulation of complex IV: the former by modulation of CoQ activity, the latter by cardiolipin-mediated ETC activation. Moreover, 2OHOA is revealed here as an inhibitor of mitochondrial β-oxidation by decreasing acylcarnitine levels and inhibiting the mitochondrial import of fatty acid β-oxidation substrates. Overall, in the face of the glycolytic impediment imposed by 2OHOA through a so far unknown mechanism, 2OHOA can also support some compensatory mitochondrial function as the sole source for ATP production, which could sustain cell viability. Thus, our results suggest that mitochondrial function is an important therapeutic target that could potentially boost the antineoplastic potential of 2OHOA.

## Figures and Tables

**Figure 1 cells-11-00578-f001:**
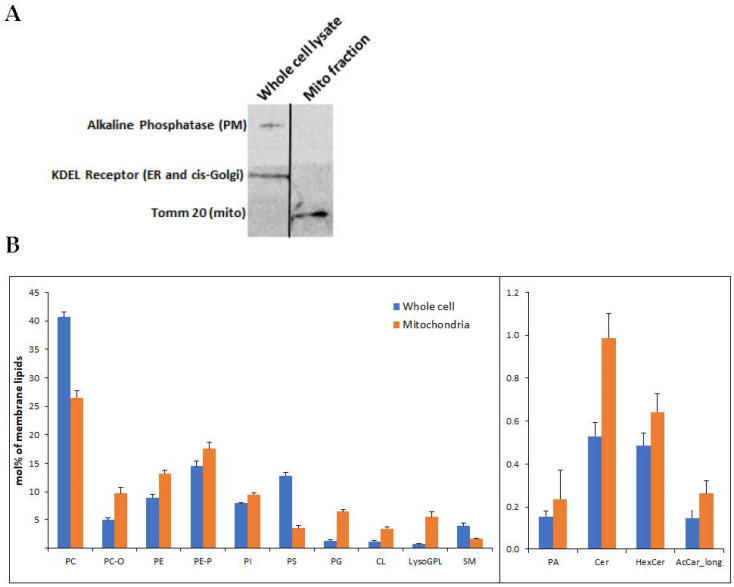
(**A**) Mitochondria were isolated from U118MG cells according to Kappler et al. The mitochondrial isolate and an identical amount (20 μg) of the whole cell lysate were subjected to SDS-PAGE and immunoblotting using antibodies against the PM marker alkaline phosphatase, the endoplasmic reticulum marker KDEL sequence and the mitochondrial marker Tom20. Representative results from 3 different experiments are shown. (**B**) Lipidomics of whole cell input and isolated mitochondrial fraction. The enrichment of CL (*p* < 3.54 × 10^−9^) and PG (*p* < 2.68 × 10^−6^) can be observed in the mitochondrial fraction, which is also depleted of several PM markers, such as PC (*p* < 2.17 × 10^−8^) and PS (*p* < 2.91 × 10^−12^). The levels of the other lipids were also different between whole cells and isolated mitochondria: PC-O (*p* < 1.47 × 10^−7^), PE (*p* < 5.45 × 10^−7^), PE-P (*p* < 0.003), PI (*p* < 4.00 × 10^−5^), LysoGPL (*p* < 0.0004), SM (*p* < 0.001), PA (*p* < 0.01), Cer (*p* < 1.51 × 10^−6^), HexCer (*p* < 0.0004), AcCar_long (*p* < 0.101). Shown are mean ± SEM, *n* = 6. Differences determined by Student’s *t*-tests. PC and PC-O, phosphatidylcholine (diacyl and alkyl-acyl); PE and PE-P, phosphatidylethanolamine (diacyl and alkenyl-acyl); PI, phosphatidylinositol; PS, phosphatidylserine; PG, phosphatidylglycerol; CL, cardiolipin; LysoGPL, Lyso glycerophospholipids; SM, sphingomyelin; PA, phosphatidic acid; Cer, ceramide; HexCer, hexosyl ceramide; AcCar_long, long-chain acylcarnitine.

**Figure 2 cells-11-00578-f002:**
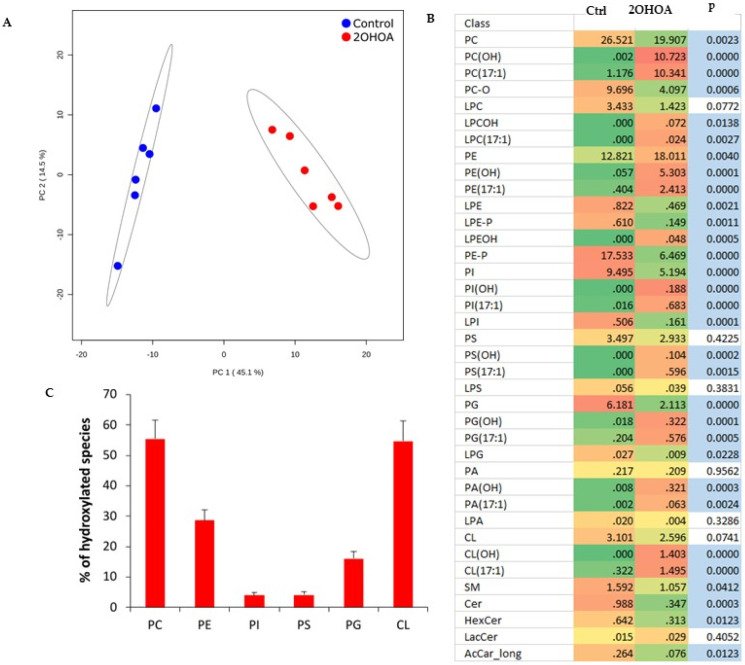
Comparative lipidomics of mitochondria isolated from U118MG cells, untreated, or treated with 200 μM 2OHOA for 48 h. *n* = 6 replicates from each treated and untreated conditions were analyzed. (**A**) Principal component analysis scores plot of lipidomic dataset ellipses displaying 95% confidence regions. (**B**) Heatmap representation of 2OHOA-induced lipidomic changes at the level of lipid classes. OH and 17:1 suffixes indicate 2OHOA- and 17:1-incorporated lipid species, respectively. Blue fill indicates *p* < 0.05 (Student’s *t*-test). (**C**) Percentage of hydroxylated species relative to non-2OHOA-generated molecules. Shown are mean ± SEM. In addition to the acronyms in Figure 1B, the “L” prefix indicates the lyso derivatives of fully acylated lipid classes. LacCer, lactosylceramide.

**Figure 3 cells-11-00578-f003:**
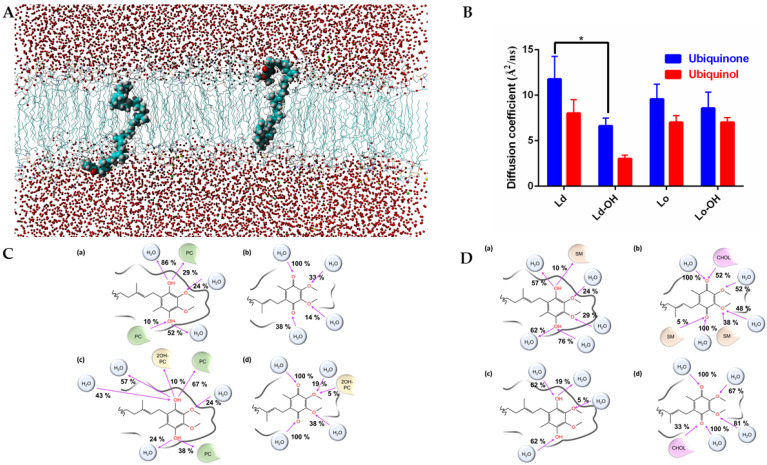
(**A**) Snapshot of the ubiquinol embedded in a POPC/2OH-POPC fully solvated membrane. (**B**) Diffusion coefficients of ubiquinone and ubiquinol in model **Lo** and **Ld** membranes. The (-OH) suffix indicates a partial hydroxylation of the lipid chain. Two-way ANOVA (*n* = 3) show significant (*p* < 0.0001) differences between the row (**Ld**, **Ld**-OH, **Lo**, **Lo**-OH) and column (ubiquinol v ubiquinone) factors. *, Student’s *t*-test shows significant difference (*p* < 0.027) between hydroxylated and non-hysdroxylated lipids in **Ld** membrane. (**C**) Graphical representation of the hydrogen bonds occurring in the last 10 ns of simulation between the ubiquinol (**a**,**c**) or ubiquinone (**b**,**d**) and the water/membrane system. In (**a**,**b**), we simulated the ubiquinol and ubiquinone embedded in a pure POPC membrane (the lipids were indicated in green as PC). In (**c**,**d**), we simulated the ubiquinol and ubiquinone embedded in a membrane model POPC/2OH-POPC (50%–50%). The hydroxylated POPC was indicated as 2OH-PC and colored in yellow. The pink arrow indicates whether the oxygen atom is an acceptor or donor of hydrogen bonds. Interactions with a frequency lower than 5% were not shown. (**D**) Same as in (**C**), but in (**a**,**b**) we simulated the ubiquinol and ubiquinone embedded in a membrane model SM/CHOL (60%–40%). The SM lipids were indicated as SM and colored in light orange. The cholesterol was indicated as CHOL and colored in pink. In (**c**,**d**) we simulated the ubiquinol and ubiquinone embedded in a membrane model SM/2OH-SM/CHOL (30%–30%–40%). The pink arrow indicates whether the oxygen atom is an acceptor or donor of hydrogen bonds. Interactions with a frequency lower than 5% were not shown. The hydroxylated form of SM shows no relevant interactions with CoQ.

**Figure 4 cells-11-00578-f004:**
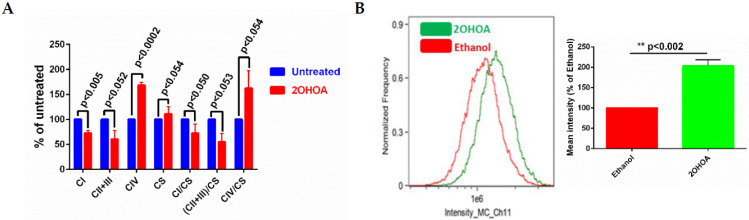
(**A**) Respiratory complex activities of control and 2OHOA (200 μM, 48 h) treated U118MG cells. In addition to respiratory complexes, the activity of citrate synthase (CS) was also measured to report the biomass of intact mitochondria. Citrate synthase normalized values are also shown. All differences between 2OHOA and untreated mitochondria are statistically significant as determined by multiple *t*-tests (*n* = 3, Holm–Sidak method, assuming no interdependence among comparisons and no consistent variance among the means). *p* values are indicated. (**B**) *Left panel*, U118MG cells were treated with 200 μM 2OHOA for 48 h, or with an equal volume of 0.5% ethanol as a solvent-only control, as indicated. Cells were then fixed and stained by Mitotracker Deep Red to quantify mitochondrial mass. *Right panel*, quantification of the differences between 0.5% ethanol and 2OHOA, based on *n* = 3 experiments as shown in the *Left panel*. Shown are means ± SEM, *n* = 3. Statistical significance (*p* value) was determined by Student’s *t*-test.

**Figure 5 cells-11-00578-f005:**
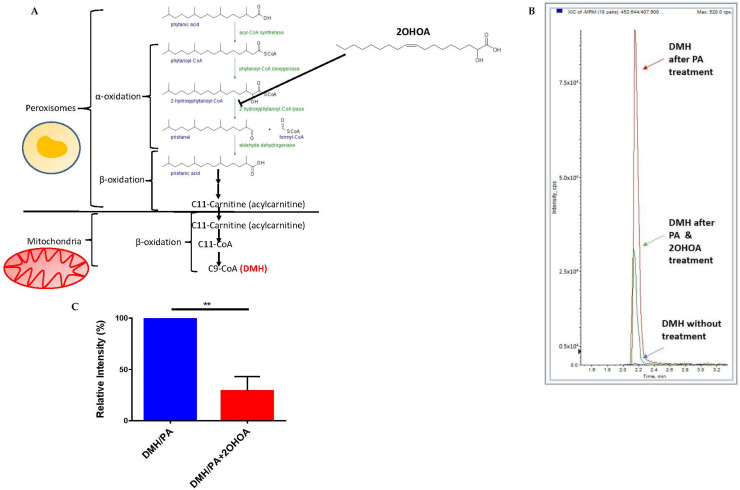
(**A**) A schematic showing the cascade from phytanic acid in peroxisomes to the mitochondrial β oxidation product C9-CoA (DMH) and the proposed involvement of 2OHOA in this cascade. (**B**) U118MG cells were treated with phytanic acid as a precursor for β oxidation of fatty acids in mitochondria (PA). The production of the specific β oxidation product dimethylheptanoyl coenzyme A (DMH) was analyzed in cells by mass spectrometry after 48 h with PA alone, PA and 200 µM 2OHOA, or without adding any substance at all. The results show a clear inhibition of β oxidation by 2OHOA, as quantified in (**C**) (*n* = 3, mean ± SEM; **, *p* < 0.01, Student’s *t*-test).

**Figure 6 cells-11-00578-f006:**
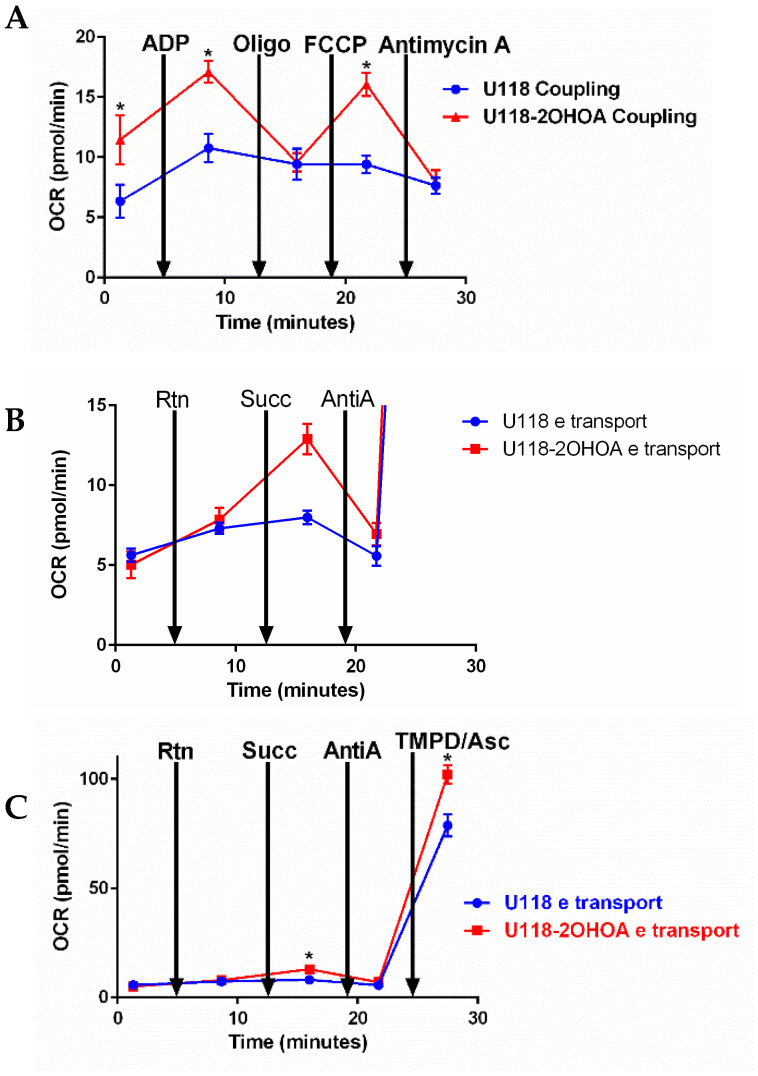
Mitochondria (2 μg/well) were isolated from untreated and 2OHOA-treated (200 μM, 48 h) U118MG cells. Respiratory activities of each complex were analyzed using Agilent’s Seahorse Mito Stress kit according to Agilent’s application note https://www.agilent.com/cs/library/applications/5991-7145EN.pdf, accessed on 4 February 2022. In coupling experiments (**A**), using rotenone and succinate as a substrate, OCR was determined following the addition of indicated reagents. An increase in OCR following ADP and FCCP supplementation indicates a unique uncoupling effect of 2OHOA. (**B**) (blowup of (**C**)) Electron flow experiments showing OCR following the supplementation of the indicated inhibitors initially using pyruvate-malate as a substrate and 4 μM FCCP to disengage electron flow from the proton motive force. Electron flow was not affected by the complex I inhibitor rotenone (Rtn). Still, it was significantly increased in 2OHOA treated cells upon supplementation of the complex II substrate succinate and decreased by supplementation of the complex III inhibitor antimycin A. Complex IV activity, measured following the supplementation of its substrate TMPD/Ascorbate, was significantly increased by 2OHOA. OCR, Oxygen consumption rate; Rtn, Rotenone; Succ, Succinate; Asc, Ascorbate. Shown are means and S.D. of *n* = 3 experiments. *, significant difference between treated and untreated (*p* < 0.05), determined by Student’s *t*-test.

**Figure 7 cells-11-00578-f007:**
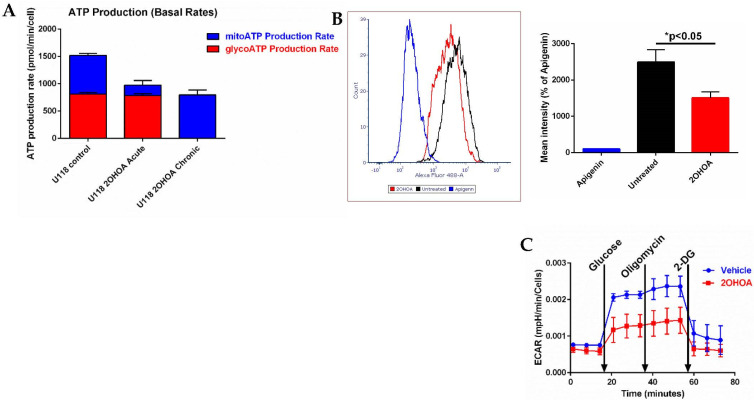
(**A**) Glycolytic (red) and mitochondrial (blue) ATP production determined by Agilent’s Seahorse machine and real-time ATP rate assay kit. U118MG cells were treated, or not (control), with 200 μM 2OHOA for 48 h (chronic), or for 20 min on assay (acute). Readings were normalized to cell numbers as determined by Crystal Violet staining. Shown are mean and SEM values based on *n* = 3 repeats. Chronic treatment with 2OHOA led to complete depletion of glycolytic ATP production, while acute treatment significantly reduced respiratory ATP production (*p* < 0.002) leaving glycolytic ATP production intact (*p* < 0.92). These differences were inferred statistically by one-way ANOVA with Sidak’s post hoc correction. (**B**) Glycolysis estimated by NBDG (2-Deoxy-2-[(7-nitro-2,1,3-benzoxadiazol-4-yl)amino]-d-glucose) uptake into U118MG cells, as quantified by flow cytometry. U118MG cells were treated with 200 μM 2OHOA for 48 h and then assayed for NBDG uptake as per the manufacturer instructions. Apigenin is a Glut 1 inhibitor and serves as a control for inhibition of NBDG uptake. Right panel shows quantification based on *n* = 3 experiments. (**C**) Glycolysis estimated by the Seahorse XF Glyco Stress kit based on *n* = 6 replicates. U118MG cells (~40,000 per well as determined by crystal violet staining) were treated, or not (vehicle), with 200 μM 2OHOA for 48 h and their glycolysis was determined by extracellular acidification rate (ECAR) using the Seahorse XF Glyco Stress kit. Basal and maximal glycolysis were respectively estimated by supplementations of 10 mM glucose and 1 μM oligomycin, while non-glycolytic ECAR was estimated by blocking glycolysis with 2-deoxyglucose (2-DG).

## Data Availability

Not applicable.

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
