# Peer review of "Multifaceted Analyses of Isolated Mitochondria Establish the Anticancer Drug 2-Hydroxyoleic Acid as an Inhibitor of Substrate Oxidation and an Activator of Complex IV-Dependent State 3 Respiration"

_cells, 2022, doi:10.3390/cells11030578_

Round 1
Reviewer 1 Report
The manuscript entitled “Multifaceted analyses of isolated mitochondria establish the anticancer drug 2-hydroxyoleic acid as an inhibitor of substrate oxidation and an activator of complex IV-dependent state 3 respiration” submitted by Kumudesh Mishra proposes how 2OHOA can have a potential anticancer activity by enhanced complex IV activity and state 3 respiration along with modulation of mitochondrial lipid composition. The manuscript has high interest in the scientific community, and the article is well-written. All the methods are explained in detail, and the experimental design is appropriate. Thus, I would be happy to recommend this manuscript for publication after addressing the following points:
- I missed the justification of why you chose the glioblastoma U118MG cells. It is not indicated in the abstract and introduction, not included in the material and methods, and only the last paragraph of the discussion covers some of these aspects. Please, add the corresponding information.
- The use of NBDG to measure glucose uptake is controversial (Immunometabolism.2020;2(4):e200029. https://doi.org/10.20900/immunometab20200029). I suggest using another methodology or at least detailing its limitations.
- According to the exciting results of ATP production, I recommend monitoring glycolytic activity by Seahorse.
- A section of statistics in Material and Methods is necessary. In addition, I missed some stats in several figures (Fig 1, 4, 6, 7).
- Figure legend: In some figures, the number of experiments is not included. Legends of figure 7: Two points at the end.
Reviewer 2 Report
The manuscript to me is, in general, clearly written. The science and technical execution of the study is of good quality. The study is solid and the data, in general, support the conclusions. The theory, logic, and experimental design are easy to follow and in general make sense. So, I recommend accepting this manuscript with minor corrections.
*Abstract is need to add some missing methods. Support the abstract with a brief methodology.
*Please make keywords different from words included in titles to increase the visibility of your work.
Although the study is original and has a clear mechanistic approach, the novelty of the work needs to be further substantiated. The study of Barnette, B.L., et al. 2021 (doi: 10.3390/ijms222111806) should be cited in the Introduction or Discussion.
Reviewer 3 Report
Mishra et al present a timely article that focuses on the use of 2OHOA in glioblastoma cells. They have presented several data that support their contentions, these include lipidomics, molecular modelling and OXPHOS activity using two approaches (spectrophotometric and oxygen consumption). The data are well controlled for the most part and are presented well. The conclusions the authors presented are mostly consistent with the data, and are timely and should advance the field considerably as the authors highlight.
Below are comments that can be used to help improve the manuscript that the authors present, most of these are minor:
- Several of the abbreviations of the fatty acids need to be made clear and fully written out, for instance, PE, DAG, PC etc. These are clear to experienced biochemists but this may be inaccessible to some readers
- The sentence starting at lines 83-84 might benefit from rewording to provide greater clarity.
- In the methods at line 94, there seems to be some misunderstanding as to what was done. The authors state an absolute amount of protein loaded in the methods, however, in the figure legend of Figure 1 the authors state 10% of the whole cell sample. This should be clarified, is 10% of 20 ug meant? Or a totally different amount.
- At line 102, the antibody should be TOMM20, perhaps the authors should consider adding the product numbers of all antibodies. This information would be important for ensuring the reproducibility of the work.
- At line 185, the authors state defatted BSA, they do not state if this is fraction V. More importantly, the authors should indicate the supplier of the BSA used.
- In section 2.7 the authors should give the concentrations of molecules used i.e. ADP, oligomycin, rotenone etc.
- At line 222, the authors note the decreased acylcarnitines, but there is no significant explanation or possible explanation of this. The authors should visit this point in the discussion.
- At line 261, the authors refer to phytanic acid in the plural, this should be singular.
- The authors should consider adding a brief schematic showing the breakdown of phytanic acid using chemical structures such that a non-expert reader can understand the point of the breakdown.
- In Figure 5, the authors should consider adding the statistical test that was performed to arrive at the significance shown. This is not in the legend at present.
- In Figure 6 at line 370, the authors should specify which Seahorse kit is used, which needs more clarity.
- At line 41, the word "from" would be better than "between" in my opinion, as the word between leaves some lack of clarity about the movement of electrons in that sentence.
- In line 498, the authors state there is inhibition of mitochondrial import of fatty acids, this I think might be the one conclusion that is not thoroughly discussed how it was arrived at.
Round 2
Reviewer 1 Report
The authors adressed all the concerns and included new data. I recommend the manuscript for publication